# Distribution of an analgesic palmitoylethanolamide and other N-acylethanolamines in human placental membranes

**Alzbeta Svobodova[1◉], Vladimir Vrkoslav[2◉], Ingrida Smeringaiova[3◉], Katerina Jirsova[3◉] ***

**1** First Faculty of Medicine, 2nd Department of Surgery–Department of Cardiovascular Surgery, Charles University and General University Hospital in Prague, Prague, Czech Republic, **2** The Institute of Organic Chemistry and Biochemistry of the Czech Academy of Sciences, Prague, Czech Republic, **3** First Faculty of Medicine, Laboratory of the Biology and Pathology of the Eye, Institute of Biology and Medical Genetics, Charles University and General University Hospital in Prague, Prague, Czech Republic

◉ These authors contributed equally to this work.

* katerina.jirsova@lf1.cuni.cz

**Data Availability Statement:** All relevant data are within the paper and its Supporting Information file.

## Abstract

### Background

Human amniotic and amniochorionic membranes (AM, ACM) represent the most often used grafts accelerating wound healing. Palmitoylethanolamide, oleoylethanolamide and anandamide are endogenous bioactive lipid molecules, generally referred as N-acylethanolamines. They express analgesic, nociceptive, neuroprotective and anti-inflammatory properties. We assessed the distribution of these lipid mediators in placental tissues, as they could participate on analgesic and wound healing effect of AM/ACM grafts.

### Methods

Seven placentas were collected after caesarean delivery and fresh samples of AM, ACM, placental disc, umbilical cord, umbilical serum and vernix caseosa, and decontaminated samples (antibiotic solution BASE 128) of AM and ACM have been prepared. Ultra-high-performance liquid chromatography-tandem mass spectrometry was used for N-acylethanolamines analysis.

### Results

N-acylethanolamines were present in all studied tissues, palmitoylethanolamide being the most abundant and the anandamide the least. For palmitoylethanolamide the maximum average concentration was detected in AM (350.33 ± 239.26 ng/g), while oleoylethanolamide and anandamide were most abundant in placenta (219.08 ± 79.42 ng/g and 30.06 ± 7.77 ng/g, respectively). Low levels of N-acylethanolamines were found in serum and vernix. A significant increase in the levels of N-acylethanolamines (3.1–3.6-fold, P < 0.001) was observed in AM when the tissues were decontaminated using antibiotic solution. The increase in decontaminated ACM was not statistically significant.

**Funding:** This work was supported by the NV18-08-00106 grant from the Ministry of Health of the Czech Republic, and by project Ministry of Education, Youth and Sports BBMRI_CZ LM2018125. Institutional support (Charles University, Prague) was provided by program Cooperatio: Medical Diagnostics and Basic Medical Sciences (KJ). The funders had no role in study design, data collection and analysis, decision to publish, or preparation of the manuscript.

**Competing interests:** The authors have declared that no competing interests exist.

## Conclusions

The presence of N-acylethanolamines, particularly palmitoylethanolamide in AM and ACM allows us to propose these lipid mediators as the likely factors responsible for the anti-hyper-algesic, but also anti-inflammatory and neuroprotective, effects of AM/ACM grafts in wound healing treatment. The increase of N-acylethanolamines levels in AM and ACM after tissue decontamination indicates that tissue processing is an important factor in maintaining the analgesic effect.

## Introduction

Profound wound healing efficiency and relatively high availability of the source tissue, the placenta, make the amniotic membrane (AM) and amniochorionic membrane (ACM) ones of the most widely used grafts worldwide. Their wound healing effect is given by the interplay of anti-inflammatory, anti-microbial, and anti-fibrotic components [1,2]. The presence of numerous growth factors, cytokines and other bioactive proteins in AM/ACM has been experimentally proven [1–5] and their therapeutic efficacy has been repeatedly confirmed at the clinical level in ophthalmology, surgery, and wound healing [6–9]. It is generally known that the difference between AM and ACM lies in the structure and biological properties. The AM is about 0.02–0.05 mm thick the innermost layer of the placenta and is composed of a single epithelial layer, a thick basement membrane and an avascular stroma. The ACM forms the outermost layer of the amniotic sac surrounding the fetus during pregnancy, while the innermost layer of this sac is the AM. The presence of stem cells (both epithelial and mesenchymal ones), biological active compounds and thick stroma make both tissues an optimal scaffold for wound healing [1–6,10].

The application of an AM/ACM graft on the wound leads to the significant, sometime even dramatic, pain relief [10–12]. To date, no specific substances directly responsible for the analgesic effect of these grafts have been reported [13,14]. The pain-relieving effect of AM/ACM is usually explained by the mechanical protection of exposed nerve endings after a tight adherence of the graft to the wound surface, hydration of the wound bed and the presence of anti-inflammatory and anti-scarring components which may alleviate nociception indirectly [13–15]. However, it seems unlikely that such a strong and generalized analgesic effect is invoked without major contribution from a specific compound [13,14]. As specific proteins have been determined to be responsible for the anti-inflammatory and anti-microbial properties of AM [1–4], we hypothesized that some active compounds should be involved also in the analgesic effect of AM/ACM. Therefore, we were interested in establishing whether some physiologically occurring substances that have proven analgesic properties are present in placental tissue, and thus explain the substantial pain relief generally reported in relation to application of AM/ACM. After an extensive search (PubMed, combination of several key words including: analgesic, nociceptive, pain, intrinsic, endogenous, growth factors, human) for potential candidates we shortlisted a group of endogenous bioactive lipid-related signaling molecules—N-acylethanolamines (NAEs), in which particularly palmitoylethanolamide (PEA) has been shown to have strong analgesic and anti-nociceptive effect [16,17].

PEA along with other endogenous fatty acid amides, oleoylethanolamide (OEA), and arachidonoylethanolamid (anandamide, AEA), are ubiquitous in organisms from plants to mammalian tissues [18–22]. In the human body, NAEs have been detected in most organs, tissues (e.g. brain, nerves, muscles, gastrointestinal tract, adipose tissue, skin, eye), and fluids (e.g.

blood, breast milk, amniotic fluid, saliva) [18–25]. All three lipid mediators (PEA, OEA, AEA) are synthesized constitutively or on demand, i.e., after exposing cells to specific, predominantly non-physiological stimuli. PEA and OEA exert their action primarily by activating the nuclear peroxisome proliferator-activated receptor-α (PPAR-α), the transient receptor potential cation channel subfamily V member 1 (TRPV1) and the G protein-coupled receptors [26–30]). AEA ligates cannabinoid receptors CB1R, CB2R, TRPV1, PPAR-γ and some evidence points to AEA binding PPAR-α also [31–33]. NAEs are implicated in multiple physiological (immunity, fertilization, feeding and sleeping behaviors) and pathological conditions (pain, inflammation, allergy) [23,24,34,35].

The anti-anaphylactic, anti-inflammatory, antifibrotic, neuroprotective, analgesic and anti-nociceptive effects of PEA were proven in experimental animal studies [16,19–21,34,36–39], and confirmed in human clinical trials [40,41]. PEA has also been shown to have a positive effect on the treatment of viral infections [36]. The anti-hyperalgesic effect of PEA has been utilized in the treatment of peripheral neuropathy and chronic pain, e.g. sciatic pain or pain from carpal tunnel syndrome [40,42,43], and its efficiency is independent of the etiopathogenesis of pain [42]. OEA has mostly anorexigenic properties [44], but its ability to reduce nociceptive responses and inflammation has also been shown [45,46]. AEA has been implicated in possessing anti-nociceptive, vasodilation, and anti-inflammatory effects [22,32,47–49] and recently its role in physiological wound healing has been suggested [22,48,49].

Binding of PEA to the PPAR-α receptor leads to attenuation of nociceptive and inflammatory responses, as confirmed in PPAR-α null mice [50,51]. The activation of PPAR-α can aid in the regeneration of mice peripheral nerves at the level of axon repair [52], PPARα activation downregulates nuclear factor kB (NF-kB) followed by the decrease of pro-inflammatory proteins, such as inducible NO synthase (iNOS), cyclo-oxygenase-2 (COX2), tumor necrosis factor-α (TNFα) or interleukin 1 and 6 or prostaglandin E2 (PGE2) [41,53], which all can contribute to the anti-inflammatory properties of AM/ACM grafts.

To date, of the NAEs lipid mediators, only AEA has been reported in placental tissues [25,54]. However, its concentrations were not measured in isolated AM or ACM [55], which are the most important placenta related tissues used for grafting. In this work we aimed to determine whether NAEs and particularly PEA, known to have strong analgesic effects, are present in placental tissues and can thus be responsible for the pain relief generally observed after AM/ACM dressing application.

## Materials and methods

### Chemicals and reagents

The standard of palmitoylethanolamide (PEA ≥ 98%) and solvent ethyl acetate (for LC-MS) were purchased from Merck (Darmstadt, Germany). The standards of oleoylethanolamide (OEA ≥ 98%), arachidonylethanolamide (AEA, MaxSpec standard quality) and palmitoylethanolamide (PEA-d4, with ≥ 99% deuterium incorporation) were obtained from Cayman Chemicals (Ann Arbor, MI, US). Acetonitrile (Optima, LC-MS grade), methanol (Optima, LC-MS grade) and formic acid (Optima LC/MS) were obtained from Fisher Scientific (Loughborough, UK). Physiological solution (0.9% w/v, Fresenius Kabi, Germany) and tissue decontamination solution BASE 128 (Alchimia srl, Italy) were used as purchased. Water was prepared using a Milli-Q integral system (Merck Millipore, Burlington, MA, USA).

### Specimen preparation

The study was approved by the Ethics Committees of the General Teaching Hospital and the First Faculty of Medicine of Charles University, Prague, Czech Republic and adhered to the

tenets set out in the Declaration of Helsinki for research involving human subjects. Human placentas were obtained after a caesarean section delivery in Motol University Hospital (Prague, Czech Republic), and the General University Hospital (Prague, Czech Republic) from donors with a normal pregnancy who signed a written informed consent. Only healthy donors (mean age 36 years), screened for hepatitis B and C, syphilis and HIV were involved. The placentas with evident pathologies or visible injuries, such as hematomas, were excluded.

Specimens were prepared from seven fresh placentas (P1 –P7), which were processed within two hours after retrieval. The tissue was repeatedly rinsed by physiological solution and blood clots were removed. Then the placental membranes (AM, ACM) were manually separated from the residual placental tissue, dissected, transferred to a mesh support (Sanatyl, Tylex Letovice, Czech Republic) and divided into 4 $cm^2$ samples [6,56]. AM and ACM were processed as either fresh tissue or decontaminated tissue (AM-d, ACM-d) with BASE 128 for 24 h at 4–8˚C, according to standard decontamination procedures used for placental membranes in clinical practice [6,56]. As concentrations of some proteins are not homogeneously distributed within placenta and AM [57], we collected samples from two different areas of placental disc (PL): from the area near the umbilical cord described as central amniotic membrane or placenta (AM1, PL1) and the area at the edge of placental disc (periphery of the placenta, AM2, PL2) [58,59]. The ACM consisting reflected amnion from mid zone of chorionic leave was used for experiments [58]. Samples of approximately 0.125 $cm^3$ were prepared from placental disc (after removing AM) and from umbilical cord (UC). All samples were prepared and then analysed in triplicates (i.e. three different samples from each donor/ tissue areas) for every specified tissue. A respective vernix caseosa (vernix, VX) sample was collected from healthy newborns immediately after full term delivery and frozen at -80˚C until processing.

Due to the finding that the decontamination increases levels of NAEs in tested tissues, we investigated the concentrations of NAEs after storage in two other solutions—Dulbecco's Modified Eagle Medium (DMEM, (c.n. 32430027, Thermo Fisher Scientific, US), usually used for storage of AM [1], and in saline (NaCl 0.9%, B Braun Melsungen AG, Melsungen, Germany). The samples were incubated at three different temperatures—4˚C, 25˚C, 37˚C—for 24 hours.

In addition, umbilical serum (US) samples were prepared from each placenta. Umbilical cord blood, which is also covered by amniotic membrane [59], was collected from fresh placenta immediately upon reception, precipitated for 2–3 hours at room temperature (RT), centrifuged (3 000 g, 15 min, 4˚C) and stored at -80˚C until NAEs measurement (two months at maximum). A list of all samples is given in Table 1.

## Specimen preparation for N-acylethanolamines detection

For analysis, all tissue samples (fresh and decontaminated) were washed in saline and homogenized mechanically by scissors for 120 seconds in 1 ml of cold acetonitrile [60]. To all homogenates, 10 μl (1 μg/ml) of PEA-d4 internal standard solution was added, and these were then allowed to shake at 4˚C and 800 rpm for 22–24 hours. Insoluble material was removed by centrifugation (20 min, 15 000 g, 4˚C); the collected extracts (900 μl) and remaining pellets were stored at -80˚C until NAEs detection (up to two months). The extracted material (pellets) was dried in an evacuated centrifuge (Refrigerated CentriVAp Concentrator, Labconco Corporation, Kansas City, MO, US) and weighed (analytical balance).

The vernix (60–75 mg) was placed into 1.5 ml Eppendorf tubes and suspended in 1.5 ml acetonitrile/ethyl acetate (2:1). The volume of 10 μl (1 μg/ml) PEA-d4 internal standard solution was added. The mixture was sonicated 30 min in ice-cold water, shaken 30 min at 4˚C

**Table 1. The list of specimens from human term placenta for NAEs analysis.**

| Specimens | Description | Size |
|---|---|---|
| AM1 | Fresh amniotic membrane from the area neighbouring the umbilical cord | 400 mm$^2$ |
| AM2 | Fresh amniotic membrane from the edge (periphery) of the placenta | 400 mm$^2$ |
| ACM | Fresh amniochorionic membrane | 400 mm$^2$ |
| AM1-d | Amniotic membrane from the area neighbouring the umbilical cord after 22-24h decontamination in BASE 128 (4°C) | 400 mm$^2$ |
| AM2-d | Amniotic membrane from the edge (periphery) of the placenta after 22-24h decontamination in BASE 128 (4°C) | 400 mm$^2$ |
| ACM-d | Amniochorionic membrane after 22 – 24h decontamination in BASE 128 (4°C) | 400 mm$^2$ |
| PL1 | Placental segment (chorionic plate and intervillous space of placenta) from the area neighbouring the umbilical cord | 225 mm$^3$ |
| PL2 | Placental segment (chorionic plate and intervillous space of placenta) from the edge (periphery) of the placenta | 225 mm$^3$ |
| UC | Umbilical cord segments from the base of umbilical cord | 225 mm$^3$ |
| US | Umbilical serum obtained from umbilical cord and placenta | 500–1000 (μl) |
| VX | Vernix caseosa from healthy newborn subjects (3 males, 4 females) delivered at full term immediately after delivery | 30–50 (mg) |

(Thermomixer, Eppendorf, Germany) and centrifuged 15 min at 16 000 g at 4°C (Centrifuge 5417R, Eppendorf). The supernatant was transferred to the new Eppendorf tube.

The extracts were evaporated to dryness in a vacuum centrifuge set at 0°C (Refrigerated CentriVAp Concentrator, Labconco) and re-dissolved in 1 ml of 30% (v/v) methanol/Milli-Q water. All extracts were purified by a slightly modified solid-phase extraction method [25] using a vacuum manifold (Agilent Technologies). The flow rate of solvent through the cartridge was pressure controlled to approximately 1 ml/min. Oasis HLB 1 cc, 30 mg cartridge (Waters) was washed using 1 ml of 100% methanol and preconditioned using 1 ml 30% (v/v) methanol. The extract was placed onto the cartridge. The cartridge was washed with 1 ml of 30% methanol, and the analytes were eluted with 100% acetonitrile to 2ml Eppendorf tube. Umbilical serum (0.25–0.5 ml) were spun at 13 000 g, enriched with 10 μl of PEA-d4 (1 μg/ml) standard solution, made up to 1 ml with deionized water, vortexed and purified, as described above.

## Liquid chromatography–Mass spectrometry analysis

The ultra-high performance liquid chromatography-tandem mass spectrometry (UHPLC/MS) system was composed of the ExionLC UHPLC AD chromatography system and the QTRAP 6500+ mass spectrometer (both Sciex, Foster City, CA, USA) equipped with electrospray ionization (ESI) ion source. The analysis was performed with VanGuard Acquity UPLC BEH C18 pre-column (2.1×5 mm, particle size 1.7 μm) connected to the analytical column Acquity UPLC BEH C18 (2.1×50 mm, particle size 1.7 μm) (both Waters). The temperature of the column and the autosampler were set at 40°C and 5°C, respectively. The volume of 5 μl of the solution was injected onto the column. Mobile phase A was water containing 2 mM ammonium acetate solution and 0.1% formic acid. Mobile phase B was acetonitrile containing 0.1% formic acid. The UHPLC gradient was programmed as follows: 0 min, 24% B; 0.3 min, 24% B; 1.5 min, 90% B; 2.5 min, 100% B, 3.5 min, 100% B then re-equilibrated at 4.5 min, 24% B and hold 24% B till 5 min; constant flow rate 700 μl/min.

ESI source operated in positive mode. The following ESI source parameters were used: curtain gas was set to 40 psi and collision gas to low; ion source was heated to 300°C; ion source gas 1 and 2 were both set to 50 psi, and ionization voltage was set to 5500 eV.

Product ions were monitored in multiple reaction monitoring (MRM) mode. The following transitions were set: quantification transitions—PEA (m/z 300.17 / 62.10), OEA (m/z 326.30 / 62.10), AEA (m/z 348.30 / 62.10) and PEA-d4 (m/z 304.17 / 61.20); confirmatory transitions, PEA (m/z 300.17 / 57.20), OEA (m/z 326.30 / 55.10), and AEA (m/z 348.30 / 91.10). Declustering and entrance potential was set to 61 V and 10 V, respectively. For other parameters, see S1 Table. Eight-point combined AEA, OEA, and PEA calibration curves spiked with internal standards PEA-d4 (10 μl of 1 μg/ml) were performed. The calibration curve was constructed for the analyte's relative signal intensity (for the area of the analyte peak divided by the peak area of the internal standard). Peaks integration, calibration curve construction and evaluation of analyte concentration were made using Analyst 1.6.3 (Sciex, Darmstadt, Germany). For tissues and vernix, the concentration was recalculated to the weight of extracted material.

### Data analysis and statistics

Each type of sample obtained from one placenta was analyzed in triplicate (analysis on three different samples from each donor/tissue areas). The resulting mean + SD was always calculated from 7 mean values calculated for each triplicate. The significance of differences in NAEs' concentrations between the fresh (control) and decontaminated samples of AM/ACM and placenta (PL1, PL2) were tested by Wilcoxon test. The descriptive statistics for each data set was calculated using R package [61]. Only data with a P-value of $\leq 0.05$ were considered statistically significant.

### Results

All three NAEs were detected in all tested samples with the exception of vernix where AEA levels were below the limit of quantitation. Average concentrations of PEA, OEA and AEA measured from all seven used placentas and their tissues are presented in Fig 1A, 1B and 1C, respectively. The measured values and statistics from all examined specimens are summarized in S2 Table.

As there was no significant difference between specimens from the central and peripheral parts of amniotic membrane (AM1 vs AM2) neither placenta (PL1 vs PL2), we merged the respective values into one set and calculated the reference levels for all NAEs in the two tissue types (AM and PL) as an average of all measured values. The same approach was used for decontaminated AM (AM-d) as no significant difference was found between central and peripheral amniotic membrane after decontamination either.

The highest concentration of PEA in placental tissues was detected in amniotic membrane, then the PEA levels gradually decreased in amniochorionic membrane, placenta, umbilical cord and vernix (listed in descending order). A significantly higher (P<0.0001) PEA concentration was measured, when the tissues were decontaminated prior the analysis (Fig 1A). This increase was independent of the sample location as it was detected both for peripheral (AM1-d, P<0.001) and central (AM2-d, P<0.001) samples, and also in the mean value of AM-d (P<0.001). The same situation was observed for AEA and OEA respectively, for significance see S2 Table.

The concentration of NAEs did not statistically differ between amniotic and amniochorionic membrane (P = 0.7433 for PEA, P = 0.7990 for OEA, P = 0.4880 for AEA). The highest concentrations for OEA and AEA were found in placenta, then the concentrations decreased from AM to ACM, AM, UC, and to vernix (Fig 1B and 1C).

However, contrary to AM, decontaminated ACM (ACM-d) did not show statistically significant increase in NAEs compared to fresh ACM (P = 0.3176, 0.0973, and 0.1280 for PEA, OEA, and AEA respectively) (Fig 1).

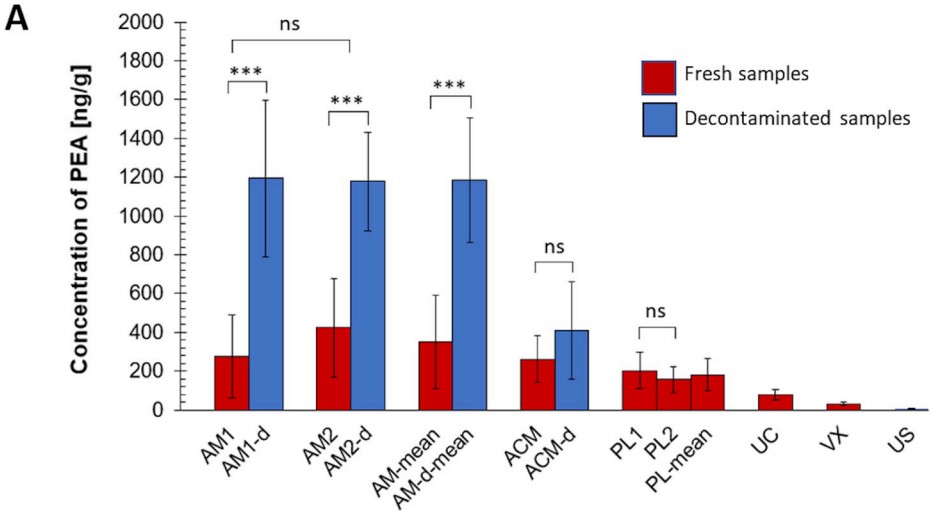

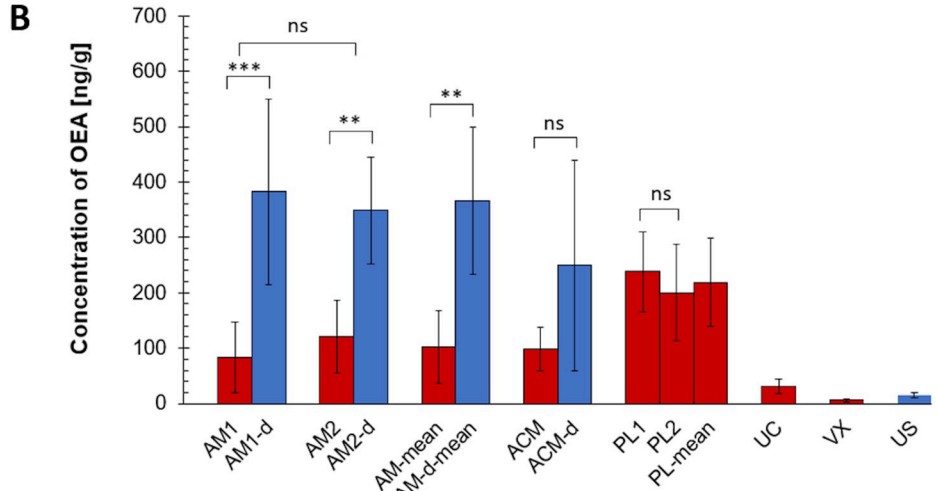

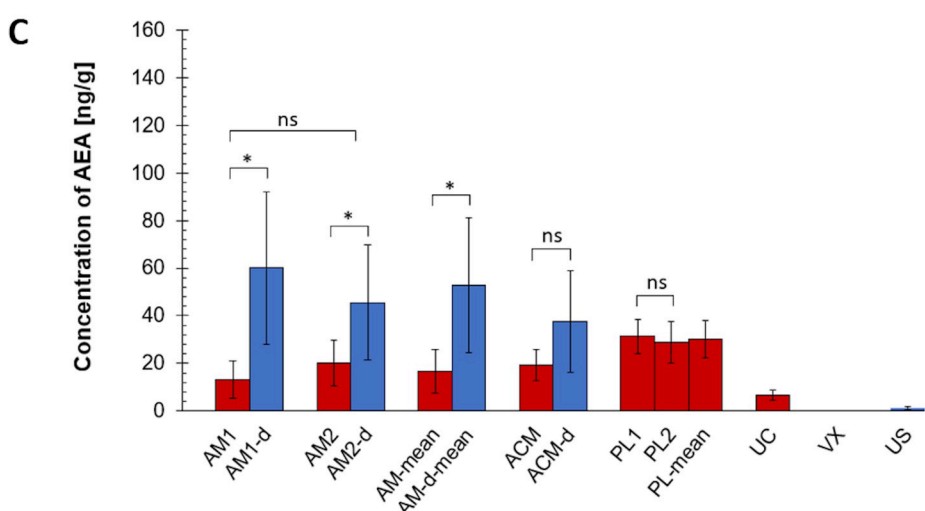

**Fig 1. The concentrations of N-acylethanolamines in various placental tissues and fluids.** Concentrations of palmitoylethanolamide (PEA) (A), oleoylethanolamide (OEA) (B), and anandamide (AEA) (C) are expressed in ng/g, except for umbilical serum (US)—in ng/ml. The average concentrations of PEA, OEA, and AEA values obtained in AM and PL samples independently of their sub-area location are also presented (AM-mean, PL-mean). P-value: $P < 0.05^*$; $P < 0.01^{**}$; $P < 0.001^{***}$; ns = non-significant; AM = amniotic membrane, ACM = amniochorionic membrane, PL = placental tissue, UC = umbilical cord, VX = vernix, d = decontaminated tissue.

To determine whether the increase in NAEs after decontamination is due to the composition of the solution or due to storage procedure, we investigated the effect of other storage solutions on NAEs levels. A comparable increase of NAEs was observed in all AM samples stored for 24 hours in culture medium or in saline (Fig 2, S3 Table).

## Discussion

In this study we demonstrated, to our knowledge for the first time, the presence of compounds with documented analgesic effect—PEA, OEA and AEA—in the placental tissues, particularly in amniotic membrane.

Because the concentrations of some proteins are known to differ in the central and peripheral part of the placenta [57], we determined the NAEs concentrations in placenta and amniotic membrane in this respect. Since no significant differences between the central and peripheral locations of AM and PL were found, we established reference concentrations of PEA, OEA and AEA in all tested tissues by averaging values from the central and peripheral part. The observed trend in the distribution of NAEs through all measured specimens (PEA > OEA > AEA) is in accordance with previously published results on various human and animal tissues [25,48]. Due to the significant proportion of lipids (9%) in the vernix caseosa [62], we were interested if NAEs are present also in this substance. The results showed that, compared to the other tested tissues, only very low levels of PEA and OEA were present

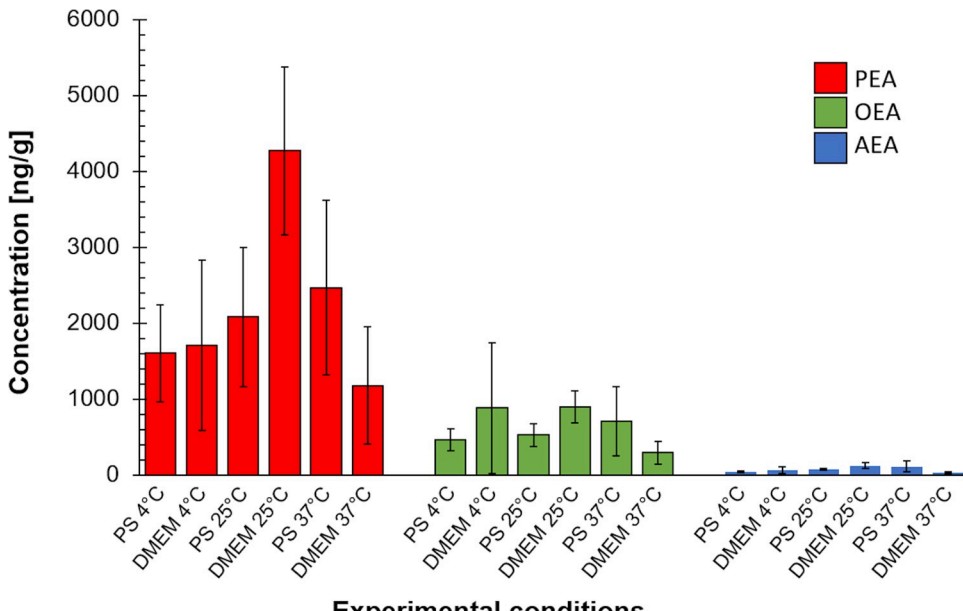

**Fig 2. Concentrations of palmitoylethanolamide (PEA), oleoylethanolamide (OEA), and anandamide (AEA) in AM samples cultured in cell culture medium (DMEM) or physiological saline (PS).** The samples were incubated at 4°C, 25°C, 37°C for 24 hours.

while AEA was not detectable. No significant difference in concentrations of PEA and OEA was found between vernix obtained from male (3) and female (4) newborns.

From placental specimens, NAEs have been previously detected in umbilical plasma and amniotic fluid [25,55], and AEA concentration has been measured in unseparated frozen placental membranes and placenta, where higher levels of AEA were found in the latter [55]. Higher concentration of AEA in placental samples compared to amniotic and amniochorionic membranes was detected also in our experiments. The same trend was observed for OEA. Opposite values (i.e. lower levels in placental samples relative to AM/ACM) were found for PEA. So far, the only determined function of placental NAEs was the stimulation of oxytocin levels by AEA in human placenta at term [54].

Our results also show that the 24-hour decontamination procedure, a necessary step in AM preparation for grafting, leads to a statistically significant increase in NAEs concentrations in AM (3.1–3.6-fold) and a slightly smaller (but not significant) elevation in ACM samples. Comparable increase in all NAEs levels in AM samples stored in culture medium or in saline indicates that the elevation of particular lipid mediators in tissue samples is more likely to be influenced by the storage duration rather than by the composition of the storage solution.

The elevation of AEA concentrations with increasing tissue processing time was previously observed by Marczylo et al. 2009 [55]. The authors also found elevated AEA levels after storage at -80˚C, which increased with the number of freeze-thaw cycles [55]. Thus, the overall NAE levels may be influenced more by tissue degradation resulting in release of lipids from tissue cells than by continuing synthesis of NAEs by surviving cells, which is balanced by cellular enzymatic degradation of these highly lipophilic, sparingly soluble compounds [63]. In any case, our results indicate that decontamination, as a basal step in AM/ACM processing, influences the quantity of NAEs in prepared grafts.

In respect of wound healing, PEA and AEA are the most interesting of the NAEs studied here because their anti-inflammatory, analgesic, anti-fibrotic and neuroprotective properties [17,34,40] strongly overlap similar features described in AM/ACM [2,3,6,56,64]. Thus, they can be at least partially responsible for the positive effects associated with AM/ACM in the healing process. We presume that the analgesic features of AM/ACM are mainly related to PEA, but anti-inflammatory and overall wound healing activities are linked also to AEA and OEA [45,46,49].

We hypothesize that PEA, released from AM after grafting operates primarily through activation of PPAR-α receptors, which are expressed in peripheral nerves and various cells recruited to the injury site including macrophages, basophiles and mast cells [34,65]. The fact that analgesic effect of PEA is suppressed in PPAR-α deficient mice [26] supports this hypothesis. PPAR-α is expressed also in the skin (thus the contact site with AM/ACM grafts) [48] where it promotes differentiation of human epidermal keratinocytes [66], and is upregulated at the edge of the wound during wound healing [67,68]. PEA-mediated activation of PPAR-α also results in the repression of pro-inflammatory NF-Kb [69], which again is profitable for wound healing. Beside that, increased expression of anti-inflammatory PPAR-γ in corneas co-cultivated with AM has been shown [70].

The detection of NAEs in AM/ACM together with above mentioned facts indicate that NAEs plays an important role in wound healing properties of AM/ACM grafts and that PEA as a NAE with the strong analgesic effect participates on the pain relief after AM/ACM application.

In future, the effect of preparation and storage conditions on the NAEs concentrations in AM/ACM should be analyzed as this is essential for the effectiveness of their application in clinical practice.

## Conclusion

In this work we detected PEA, OEA, and AEA in placental tissues and determined their basal concentrations levels. To our knowledge this is the first prove of presence of compounds with analgesic and nociceptive properties in placenta. We propose that analgesic, anti-inflammatory and neuroprotective properties of NAEs are involved in analgesic and anti-inflammatory activity of AM/ACM membrane, and substantially contribute to their wound healing effect. The increase of NAEs levels particularly in AM after a 24-hour decontamination period in antibiotic solution, indicate that the tissue processing may play an important role in maintaining or modulating the analgesic effect of the tissue prepared for grafting.

## Supporting information

**S1 Table. Parameters for the mass spectrometry detection.**
(DOCX)

**S2 Table. Concentrations of PEA, OEA, and AEA (ng/g; ng/ml for US) in various placental specimens—original data and statistics (AV, average; SD, standard deviation, values of triplicates).** The descriptive statistics for each data set was calculated using R package. Only data with a P-value of $\leq 0.05$ were considered statistically significant.
(DOCX)

**S3 Table. NAEs concentrations in AM samples stored in two additional solutions: standard cell culture medium (DMEM) or physiological saline (PS), at three different temperatures: 4˚C, room temperature (25˚C), 37˚C.** AV, average; SD, standard deviation; PL, placenta.
(DOCX)

## Acknowledgments

The authors thank MSc. Simona Krausova for help with the sample preparation, Dr. Jan Bednar for help with statistics, and Dr. Joao Victor Cabral, Dr. Catherine Joan Jackson, and Assoc. Prof. Jan Bednar for proofreading the manuscript and making language corrections.

## Author Contributions

**Conceptualization:** Vladimir Vrkoslav, Ingrida Smeringaiova, Katerina Jirsova.

**Data curation:** Alzbeta Svobodova, Vladimir Vrkoslav, Ingrida Smeringaiova.

**Formal analysis:** Vladimir Vrkoslav.

**Funding acquisition:** Katerina Jirsova.

**Investigation:** Alzbeta Svobodova, Vladimir Vrkoslav, Ingrida Smeringaiova, Katerina Jirsova.

**Methodology:** Vladimir Vrkoslav, Ingrida Smeringaiova, Katerina Jirsova.

**Project administration:** Katerina Jirsova.

**Supervision:** Katerina Jirsova.

**Writing – original draft:** Alzbeta Svobodova, Vladimir Vrkoslav, Ingrida Smeringaiova, Katerina Jirsova.

**Writing – review & editing:** Alzbeta Svobodova, Vladimir Vrkoslav, Katerina Jirsova.

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
