## [Decision Letter · Decision Letter 0]

29 Mar 2022

PONE-D-21-41101Distribution of an analgesic palmitoylethanolamide and other N-acylethanolamines in human placental membranesPLOS ONE

Dear Dr. Jirsova,

Thank you for submitting your manuscript to PLOS ONE. After careful consideration, we feel that it has merit but does not fully meet PLOS ONE’s publication criteria as it currently stands. Therefore, we invite you to submit a revised version of the manuscript that addresses the points raised during the review process.

For best readability, please convert the table 2 and the supplemental table 3 in graphs (for example histograms). Include the figures from supplemental table 3 as new figures of the paper, because these data may be useful and inherent. 

Please respond accurately to each point of Reviewer 1.

We look forward to receiving your revised manuscript.

Kind regards,

Fabio Sallustio, PhD

Academic Editor

PLOS ONE

Journal Requirements:

2. Please provide additional details regarding participant consent. In the Methods section, please ensure that you have specified (1) whether consent was informed and (2) what type you obtained (for instance, written or verbal). If your study included minors, state whether you obtained consent from parents or guardians. If the need for consent was waived by the ethics committee, please include this information.

Additional Editor Comments:

For best readability, Please convert the table 2 and the supplemental table 3 in graphs (for example histograms). Include the figures from supplemental table 3 as new figures of the paper, because these data may be useful and inherent.

Please respond accurately to each point of reviewer 1.

Reviewers' comments:

Reviewer's Responses to Questions

**Comments to the Author**

1. Is the manuscript technically sound, and do the data support the conclusions?

Reviewer #1: Yes

Reviewer #2: Yes

2. Has the statistical analysis been performed appropriately and rigorously? 

Reviewer #1: No

Reviewer #2: Yes

3. Have the authors made all data underlying the findings in their manuscript fully available?

Reviewer #1: Yes

Reviewer #2: Yes

4. Is the manuscript presented in an intelligible fashion and written in standard English?

Reviewer #1: No

Reviewer #2: Yes

5. Review Comments to the Author

Reviewer #1: The submitted study offers a direct measurement of N-acylethanolamines (natural pain killers) in different part of full-term human placentae. The authors performed ultra-HPLC in tandem with MS to determine and quantify bioactive lipids with analgesic activity in human placental tissues. The results are of particular interest to current and future clinical activities. However, the study is quite preliminary and limited in the quality and quantity of measurement.

The analgesic effect offered by human placenta, and amnion membrane in particular, has been previously described and recently tested in monocentric trial (Mohseni F et al., Crescent Journal of Medical and Biological Sciences 2018). The study performed by Svobodova and co-authors should be considered as short communication since the authors limitedly measured presence of three selected NAE compounds (PEA, OEA and AEA) in 7 specimens.

The effects of such lipid mediators have been proposed and hypothesized (by the authors and others) related to their action on cannabinoid or PPAR receptors. Unfortunately, none of such hypotheses and pathways have been further elucidated or investigated.

Furthermore, experimental settings is sometime confused and poorly described:

What’s the difference between AM and ACM?

Please offer better description (and supportive images) for the “different areas of placental disc” (at page 7 described as “the area near the umbilical cord (AM1, PL1) and the area in the periphery of the placenta”). Are the authors referring to the main placenta body (containing decidua and villi) and reflected amnion/chorion surrounding the baby?

Even the measurements on umbilical cord extracts and UC blood are quite poor and deserve additional profiling and characterization. What about amniotic fluid? why the authors did not include such additional tissue (currently used in clinical setting)?

Please motivate the comparison with selected fetal tissue (Vernix caseosa)

Introduction is particularly long and contains several information poorly or no pertinent with the current study. Discussion (above all first page) consists in repetition of the information detailed in the Introduction (please revise and erase)

In Material and Method section, Statistic paragraph, the authors referenced “descriptive statistics for each data set was calculated using R package” (ref. nr.67). Please motivate. In the same paragraph, the authors stated as “The significance of differences in NAEs’ concentrations between the fresh (control) and decontaminated samples of AM/ACM and placenta (PL1, PL2) were tested by Wilcoxon test”. Please motivate both the use of Wilcoxon and the comparison between selected groups (instead of ANOVA or similar).

Table 2 and Figure 1 can be combined.

Discussion is quite extended, and several speculations are included, considering the extremely limited amount of presented data

Reviewer #2: I found this Research paper well written.

Experimental plan is well performed.

Discussion is clear.

Materials and methods section ensures reproducibility.

As limitation, the specific topic could be interested only for very specific readers.

6. PLOS authors have the option to publish the peer review history of their article (what does this mean?). If published, this will include your full peer review and any attached files.

Reviewer #1: No

Reviewer #2: No

---

## [Author Response · Author response to Decision Letter 0]

6 Jun 2022

We duly considered all comments and suggestions of the reviewers, and we made the appropriate corrections and amendments. We appreciated all reviewers' suggestions, and we believe that they contributed to the improvement of the manuscript. All changes that we made in the manuscript are highlighted in yellow.

To the comments of the editor:

1. For best readability, Please convert the table 2 and the supplemental table 3 in graphs (for example histograms). Include the figures from supplemental table 3 as new figures of the paper, because these data may be useful and inherent.

Please respond accurately to each point of reviewer 1.

Table 2 and supplemental Table 3 are now presented as histograms (Fig.1 and Fig. 2, respectively) as editor suggested. 

Reviewers' comments:

Reviewer #1: 

1.The submitted study offers a direct measurement of N-acylethanolamines (natural pain killers) in different part of full-term human placentae. The authors performed ultra-HPLC in tandem with MS to determine and quantify bioactive lipids with analgesic activity in human placental tissues. The results are of particular interest to current and future clinical activities. However, the study is quite preliminary and limited in the quality and quantity of measurement. The analgesic effect offered by human placenta, and amnion membrane in particular, has been previously described and recently tested in monocentric trial (Mohseni F et al., Crescent Journal of Medical and Biological Sciences 2018). The study performed by Svobodova and co-authors should be considered as short communication since the authors limitedly measured presence of three selected NAE compounds (PEA, OEA and AEA) in 7 specimens.

The analgesic effect of placenta was repeatedly investigated and evaluated (PMID: 32832040, 32410269, 30559055, study by Mosheni et al., and also in our study: PMID: 34791774), on the other hand the mechanism of this analgesic effect was not related to effector molecules (see Introduction, second paragraph). We believe that triplicates from 7 placentas provide a sufficiently representative set in the terms of study reproducibility. This is evidenced by practically identical results (averages) from the area of AM1 and AM2 and PL1 and PL2, respectively.

2. The effects of such lipid mediators have been proposed and hypothesized (by the authors and others) related to their action on cannabinoid or PPAR receptors. Unfortunately, none of such hypotheses and pathways have been further elucidated or investigated.

Yes, this is true, but regarding human study, we have no idea, how to confirm clinically that NAEs are responsible for analgesic action documented after the application of AM to patient’s wound. We would like to mention, that we also detected PPAR receptor in AM, but these results will be included in the following study. Beside that, the presence of PPAR receptor in the amniotic membrane does not indicate directly that reaction between PEA and PPAR is responsible for analgesic effect of the amniotic membrane.

3. Furthermore, experimental settings is sometime confused and poorly described:

What’s the difference between AM and ACM?

It is generally known that the difference between AM and ACM lies in the structure and biological properties. The following text has been added to the Introduction section:

The AM is about 0.02-0.05 mm mm thick the innermost layer of the placenta, and is composed of a single epithelial layer, a thick basement membrane and an avascular stroma. The ACM forms the outermost layer of the amniotic sac that surrounds the fetus during pregnancy, while the innermost layer of this sac is the AM. The presence of stem cells, biological active compounds and thick stroma make both tissues an optimal scaffold for wound healing. 

4. Please offer better description (and supportive images) for the “different areas of placental disc” (at page 7 described as “the area near the umbilical cord (AM1, PL1) and the area in the periphery of the placenta”). Are the authors referring to the main placenta body (containing decidua and villi) and reflected amnion/chorion surrounding the baby?

The description of the areas the tissue was obtained from, was described in the detail, proper references were also added to M and M section:

As concentrations of some proteins are not homogeneously distributed within placenta and AM [65], we collected samples from two different areas of placental disc (PL): from the area near the umbilical cord described as central amniotic membrane or placenta (AM1, PL1) and the area at the edge of placental disc (periphery of the placenta, AM2, PL2) (Centurione 2018, PMID: 29562779, Weidinger 2021, 33520964). The ACM (amniochorionic membrane) consisting reflected amnion from mid zone of chorionic leave was used for experiments (Centurione 2018, PMID: 29562779). The particular locations of AM/ACM are well schematized in a recent paper, which has been included as one of the references (Weidinger 2021, 33520964).

5. Even the measurements on umbilical cord extracts and UC blood are quite poor and deserve additional profiling and characterization. What about amniotic fluid? why the authors did not include such additional tissue (currently used in clinical setting)?

The selection of the monitored tissues was based not only on the anatomical point of view (placenta derivatives), but also on the potential importance of the lipids in the tissue. This means that we have focused mainly on tissues that are used clinically for wound treatment and have a strong analgesic effect, i.e. AM and ACM. The levels of amniotic fluid were already assessed (see ref. Discussion, third paragraph). In our experiments, the amniotic fluid was not examined as we could focus just on tissue which is available as a surplus tissue at standard caesarean section. Additionally, the information that UC is covered by AM was included in M and M section. 

6. Please motivate the comparison with selected fetal tissue (Vernix caseosa).

Vernix caseosa (VC) is a complex biofilm composed of water in hydrated corneocytes (80%), surrounded by a matrix of lipids (10%) and proteins (10%) (PMID: 18489296, 19881987). The lipid fraction is extremely rich and not fully characterized yet, despite the efforts of numerous researches (PMID: 16628195, 28576934). Our motivations were as follows: The discovery of high concentrations of compounds could indicate the analgesic effects of vernix caseosa. Concentration levels of NAEs can also indicate, where NAEs are biosynthesized. (If they are transported to the placenta from the fetal skin.) As it is well known that vernix caseosa protects newborn baby and has some protective roles during fetal development and for first hours after birth, we were interested if in our study detected NAEs are present in this substance. The information about high level of lipids in VC was added to Discussion section.

7. Introduction is particularly long and contains several information poorly or no pertinent with the current study. Discussion (above all first page) consists in repetition of the information detailed in the Introduction (please revise and erase).

The Introduction and Discussion were shortened in line with the reviewer's requirements, the other parts required by reviewers (see above) were included.

8. In Material and Method section, Statistic paragraph, the authors referenced “descriptive statistics for each data set was calculated using R package” (ref. nr.67). Please motivate. In the same paragraph, the authors stated as “The significance of differences in NAEs’ concentrations between the fresh (control) and decontaminated samples of AM/ACM and placenta (PL1, PL2) were tested by Wilcoxon test”. Please motivate both the use of Wilcoxon and the comparison between selected groups (instead of ANOVA or similar).

Rationales (reasoning) for using Wilcoxon test:

In this study we performed only pair-wise comparison of data sets (AM fresh vs AM dried or PL1 vs PL2). Therefore, multiparametric approaches as ANOVA analysis are not appropriate. As the data sets are quite restricted the normality of the distribution is not possible to confirm (therefore the classical student t-test can be objected) and thus the Wilcoxon test (paired) is the most appropriate to obtain the information about the statistical significances of the differences of the data sets.

Table 2 and Figure 1 can be combined.

Table 2 and Figure 1 were combined as suggested by reviewer.

Discussion is quite extended, and several speculations are included, considering the extremely limited amount of presented data

Discussion was shortened complying with the reviewer's requirements.

Reviewer #2: I found this Research paper well written.

Experimental plan is well performed.

Discussion is clear.

Materials and methods section ensures reproducibility.

As limitation, the specific topic could be interested only for very specific readers.

---

## [Decision Letter · Decision Letter 1]

24 Jun 2022

PONE-D-21-41101R1

Distribution of an analgesic palmitoylethanolamide and other N-acylethanolamines in human placental membranes

PLOS ONE

Dear Dr. Jirsova,

Thank you for submitting your manuscript to PLOS ONE. After careful consideration, we have decided that your manuscript does not meet our criteria for publication and must therefore be rejected.

 The proposed study has a great potential and may be extremely interesting if properly conducted and completed with analysis and tests but at moment the described results are extremely limited and have not been improved and increased with the revisions.

I am sorry that we cannot be more positive on this occasion, but hope that you appreciate the reasons for this decision.

Kind regards,

Fabio Sallustio, PhD

Academic Editor

PLOS ONE

Reviewers' comments:

Reviewer's Responses to Questions

**Comments to the Author**

1. If the authors have adequately addressed your comments raised in a previous round of review and you feel that this manuscript is now acceptable for publication, you may indicate that here to bypass the “Comments to the Author” section, enter your conflict of interest statement in the “Confidential to Editor” section, and submit your "Accept" recommendation.

Reviewer #1: (No Response)

Reviewer #2: All comments have been addressed

2. Is the manuscript technically sound, and do the data support the conclusions?

Reviewer #1: Partly

Reviewer #2: Yes

3. Has the statistical analysis been performed appropriately and rigorously? 

Reviewer #1: No

Reviewer #2: Yes

4. Have the authors made all data underlying the findings in their manuscript fully available?

Reviewer #1: No

Reviewer #2: Yes

5. Is the manuscript presented in an intelligible fashion and written in standard English?

Reviewer #1: Yes

Reviewer #2: Yes

6. Review Comments to the Author

Reviewer #1: The R1 manuscript contains very little to null changes or improvement. The authors denied most of the changes or addition recommended or requested by reviewer.

The proposed study would have a great potential and may be extremely interesting if properly conducted and completed with analysis and tests. The described results are extremely limited and may serve for a short communication in a scientific meeting, but definitely not for a full research article

The staring hypothesis may sound, but the authors should clarify why and how they selected N-acylethanolamines as critical mediators for pain relieving effect offered by amnion/amniochorion membranes (at page 4, the authors briefly stated “After an extensive search for potential candidates we shortlisted a group of endogenous bioactive lipid-related signalling molecules - N-acylethanolamines (NAEs)”). Such statement requires additional information or complete set of analysis in support, since as the authors correctly stated PEA. OEA and AEA “are ubiquitous in organisms from plants to mammalian tissues”.

The introduction is quite detailed and contains several information not strictly pertinent with the study. The authors may consider converging such info (and more) into an interesting review manuscript on PEA and perinatal tissues

The provided short description of amnion and chorionic membranes is quite superficial. The authors should elucidate the cell population contained in both tissues and the isolation procedure for both tissutal products. Particularly, the authors should clarify their statement regarding stem cells: do they refer to amnion epithelial or mesenchymal stromal cells? Both of them has been frequently called “stem” due to peculiar properties and capacity. However amnion and chorion contains different cells. Different in genesis and biological properties.

The role of cannabinoid receptors in perinatal derivative activities is extremely interesting, with particular attention to PPAR molecules. The authors mentioned some studies ongoing on PPAR-alpha. Such results have been intentionally excluded from the current study and intended for another separate publication. We would strongly encourage authors to reconsider such decision, considering the paucity of results here presented that jeopardize possibility to publish current manuscript. Interestingly, the expression of nuclear factor-κB (NF/kB) or PPAR-γ related proteins has been previously reported in amnion membrane and underlying tissues (Bauer et al. Invest Ophthalmol Vis Sci 2012;53(2):799; Antoine et al. Life 2022;12:544).

The two areas the authors collected samples have been described with some (minor? Relevant?) properties (ref.58). No difference has also been reported by the authors between peri-umbilical and distal zone of the placental body. Surprisingly, the authors found no relevant peeling amnion out from the reflect regions to complete the analysis. AM vs ACM proteins may be extremely relevant for the proposed medical applications

Another interesting aspect of the proposed analysis is gender difference: the chemical variability of lipids contained in the vernix caseosa has been reported to be gender-specific (i.e., female newborns have apparently more wax esters and triacylglycerols with longer hydrocarbon chains). Similar correlations between male and female donor fetal tissues should be analyzed and compared.

Have the authors combined all 7 donors or analysed them separately? Most likely the first option, since the authors reported “All three NAEs were detected in all tested samples”. If so, please report all the results and concentrations

The authors analysed decontaminated tissue ability to re-express NAE levels. Such in vitro conditions should be applied also to non-decontaminated samples to properly evaluated modifications due to ex vivo conditions

In the reviewers’ rebuttal document, the authors claimed as “triplicates from 7 placentas provide a sufficiently representative set in the terms of study reproducibility”. It is not clear if the authors referred to triplicate in the running analysis (used to evaluate procedural errors) or triplicate means analysis on three different samples from every donor (and donor areas).

Statistical analysis: I personally disagree with authors statement finding ANOVA not relevant. All the perinatal compartments have been compared to each other, thus multiparametric analysis may be more relevant than t test. Distribution can only be evaluated enlarging the number of analysed samples

Reviewer #2: No further comments.

Authors well addressed my previous comments. This version of the manuscript is suitable for publication.

7. PLOS authors have the option to publish the peer review history of their article (what does this mean?). If published, this will include your full peer review and any attached files.

Reviewer #1: No

Reviewer #2: **Yes: **Dario Siniscalco

- - - - -

---

## [Author Response · Author response to Decision Letter 1]

28 Jul 2022

Reviewer #1: 

The R1 manuscript contains very little to null changes or improvement. The authors denied most of the changes or addition recommended or requested by reviewer.

The proposed study would have a great potential and may be extremely interesting if properly conducted and completed with analysis and tests. The described results are extremely limited and may serve for a short communication in a scientific meeting, but definitely not for a full research article

Our reply: 

We do not agree with this statement. In its previous comments, the referee #1 raised 10 points (see below the joint “Revision 1 - Response to Reviewers”). The comments 3, 4, 6, 7, 9, and 10 were reflected by the modification of the text. The remaining comments were in the form of general criticism (comments 1, 2, and 5), to which we gave our best explanation; see below. In comments 6 and 8, the referee asks for “motivation”. In our responses, we tried to explain our reasoning for selecting the experimental or evaluation strategy.

The staring hypothesis may sound, but the authors should clarify why and how they selected N-acylethanolamines as critical mediators for pain relieving effect offered by amnion/amniochorion membranes (at page 4, the authors briefly stated "After an extensive search for potential candidates we shortlisted a group of endogenous bioactive lipid-related signaling molecules - N-acylethanolamines (NAEs)"). Such statement requires additional information or complete set of analysis in support, since as the authors correctly stated PEA. OEA and AEA "are ubiquitous in organisms from plants to mammalian tissues". 

Our reply: 

We selected these substances as they are known for being the most natural and abundant analgesic factors in living organisms. As clearly noted in the manuscript, if their presence can also be detected in AM (which has not been accomplished before) this would be the first detection of compounds with analgesic properties in AM giving at least partial explanation of the analgesic effect when AM is applied on the wound. This fact – that the main objective of this manuscript is to detect analgesic compounds that could be responsible for AM grafts’ analgesic features – is repeatedly stated in the abstract and the introduction. Additional clarification regarding the search for analgesic agents has been added to the text (see in bold): “Therefore, we were interested in establishing whether some known physiologically occurring substances that have proven analgesic properties are present in placental tissue, and thus explain the substantial pain relief generally reported in relation to the application of AM/ACM. After an extensive search (Pubmed, combination of several keywords including: analgesic, nociceptive, pain, intrinsic, endogenous, growth factors, human) for potential candidates we shortlisted a group of endogenous bioactive lipid-related signaling molecules N-acylethanolamines (NAEs), in which particularly palmitoylethanolamide (PEA) has been shown to have profound analgesic and anti-nociceptive effect [16, 17].”

The introduction is quite detailed and contains several information not strictly pertinent with the study. The authors may consider converging such info (and more) into an interesting review manuscript on PEA and perinatal tissues. 

Our reply: Based on the demand of Reviewer #1, we removed in Revision No 1 the following parts from the Introduction:

• “They are released from cell membrane phospholipid precursors by phospholipase D, and then act locally in the cells to which they are transported by carrier-mediated transport [22, 26, 27].”

• The detailed information on NAEs receptors “…GPR55 and GPR119 [28-32]. PEA can indirectly activate cannabinoid receptors CB2R [33]. ” 

• “PEA was first isolated from egg yolk, soybeans, and peanuts where its anti-anaphylactic activity was also reported [18, 19”

• “It has been reported in rodent models that PEA inhibits lung, liver and retinal fibrosis [42-44]” 

• “(PEA)…and was recently proposed as a promising nutraceutical in COVID-19 infection [46].”

• “PEA also stimulates macrophages to remove invading bacteria and apoptotic neutrophils [60, 61]”

We understand that Reviewer #1 may not have noticed these deleted manuscript parts. Only changes and added text were highlighted in yellow (with R1 cover letter). The reason for tracking changes was to keep the manuscript readable after extensive modifications, i.e., also changes in graphs and tables. 

The provided short description of amnion and chorionic membranes is quite superficial. The authors should elucidate the cell population contained in both tissues and the isolation procedure for both tissutal products. Particularly, the authors should clarify their statement regarding stem cells: do they refer to amnion epithelial or mesenchymal stromal cells? Both of them has been frequently called "stem" due to peculiar properties and capacity. However amnion and chorion contains different cells. Different in genesis and biological properties.

Our reply: 

It is true that our response was short, but it was done based on Reviewer #1 requirement: please see below Revision 1 - Response to Reviewers, point 3. Five lines of text have been added in total. Since Reviewer #1 wanted to shorten the introduction, we did not dare to extend it with further details about amniotic membrane (AM). In addition, due to the content of the article, we perceive that most potential readers (studying or applying AM) have knowledge about the properties of AM, but they do not have information about NAEs and their properties and their relation to AM. That was the main reason why we included extensive information about NAEs and a less detailed one about AM. Regarding stem cells, the following information was added: “The presence of stem cells (both epithelial and mesenchymal stromal ones), biological active compounds and thick stroma make both tissues an optimal scaffold for wound healing [1-6, 10]. “ 

The role of cannabinoid receptors in perinatal derivative activities is extremely interesting, with particular attention to PPAR molecules. The authors mentioned some studies ongoing on PPAR-alpha. Such results have been intentionally excluded from the current study and intended for another separate publication. We would strongly encourage authors to reconsider such decision, considering the paucity of results here presented that jeopardize possibility to publish current manuscript. Interestingly, the expression of nuclear factor-κB (NF/kB) or PPAR-γ related proteins has been previously reported in amnion membrane and underlying tissues (Bauer et al. Invest Ophthalmol Vis Sci2012;53(2):799; Antoine et al. Life 2022;12:544). The two areas the authors collected samples have been described with some (minor? Relevant?) properties (ref.58). No difference has also been reported by the authors between peri-umbilical and distal zone of the placental body. Surprisingly, the authors found no relevant peeling amnion out from the reflect regions to complete the analysis. AM vs ACM proteins may be extremely relevant for the proposed medical applications.

Our reply: 

Again, this question was not raised in its first review of Reviewer #1. In these comments, the reviewer’s remark considering the additional citations, is correct and following sentence was added to the Discussion section: “Beside that, increased expression of anti-inflammatory PPAR-γ in corneas co-cultivated with AM has been shown (Bauer et al. 2012)”. The article on the influence of phthalates on the decrease PPAR-γ activity in fetal membranes (Antoine et al. 2022) was published 4 months after we submitted the first version of our manuscript to PLOS. This (mentioned) article also shows the growing importance of NAEs related to placental tissue and we would like our findings on the analgesic properties of AM related to PEA and other NAEs to be published after more than half of year effort. It should be stressed here that the selection of experiments and AM regions was a kind of “proof of hypothesis” and not a thorough full quantitative analysis of individual AM regions. At the same time, the reviewer states that the “AM vs ACM proteins may be extremely relevant for the proposed medical applications”, and this type of information is provided for NAEs in our manuscript.

Another interesting aspect of the proposed analysis is gender difference: the chemical variability of lipids contained in the vernix caseosa has been reported to be gender-specific (i.e., female newborns have apparently more wax esters and triacylglycerols with longer hydrocarbon chains). Similar correlations between male and female donor fetal tissues should be analyzed and compared. 

Our reply: 

This question was not raised in the first review of Reviewer #1; thus, we did not have an opportunity to reply to it. Yes, reviewer´s comment is correct. However, the scope of our experiments was to detect and quantify the NAEs in the AM. The vernix (VX) information was included just to show the magnitude of NEAs quantity in AM. The data on gender and significance was added to Material and Methods: “Vernix caseosa from healthy newborn subjects (3 males, 4 females) delivered at full term immediately after delivery”, and to Discussion sections: “No significant difference in concentrations of PEA and OEA was found between vernix obtained from male (3) and female (4) newborns.” Additionally, the concentrations of PEA and OEA in VX are very low (compared to AM, ACM, and placenta (11-17 times for PEA). AEA levels are below the limit of quantitation, please see Results.

Have the authors combined all 7 donors or analysed them separately? Most likely the first option, since the authors reported "All three NAEs were detected in all tested samples". If so, please report all the results and concentrations 

Our reply: 

This information was already present (since the first version) in the main text (donor combination). The values of individual triplicates were not included in the first revision as they were not requested by either of the referees in the first-round of comments. Now, these values have been added to Supporting S2 Table (yellow highlighted).

The authors analysed decontaminated tissue ability to re-express NAE levels. Such in vitro conditions should be applied also to non-decontaminated samples to properly evaluated modifications due to ex vivo conditions. In the reviewers' rebuttal document, the authors claimed as "triplicates from 7 placentas provide a sufficiently representative set in the terms of study reproducibility". It is not clear if the authors referred to triplicate in the running analysis (used to evaluate procedural errors) or triplicate means analysis on three different samples from every donor (and tissue areas). 

Our reply: 

The reviewer is correct, the explanation concerning the triplicates was incomplete, thus ambiguous. We are sorry for that and the triplicates means “analysis on three different samples from each donor/donor areas” This information had been added to the new version of manuscript.

Statistical analysis: I personally disagree with authors statement finding ANOVA not relevant. All the perinatal compartments have been compared to each other, thus multiparametric analysis may be more relevant than t test. Distribution can only be evaluated enlarging the number of analysed samples to 

Our reply: 

Unfortunately, this remark does not seem to be correct. We did not compare “all the perinatal compartments to each other”; we only measured the average concentration of NAEs in each of them. Nowhere in the text were these values statistically compared, as we did not find it relevant. On the other hand, we wanted to evaluate the processing effect on the NAEs concentration. Thus, statistical evaluations for differences were made between fresh and decontaminated tissues separately for each tissue. Therefore AM vs. AM-d, AM1 vs. AM1-d, AM2 vs. AM2-d, ACM vs. ACM-d, and PL1vsPL2 were evaluated. These are two-set comparisons, and thus the multiparametric test is not relevant. Moreover, we did not use the t-test, but the Wilcoxon test as the distribution of individual sets was not normal.

Reviewer #2: No further comments.

---

## [Editor Report · Decision Letter 2]

8 Nov 2022

PONE-D-21-41101R2

Distribution of an analgesic palmitoylethanolamide and other N-acylethanolamines in human placental membranes

PLOS ONE

Dear Dr. Jirsova,

Thank you for submitting your manuscript to PLOS ONE. After careful consideration, we feel that it has merit and we would reconsider its publication in PLOS ONE. Therefore, we invite you to submit a revised version of the manuscript, starting from your last version and your last rebuttal letter, that addresses the points raised during the review process.

Please, consider, if possible, to add new data on PPAR-alpha.

We look forward to receiving your revised manuscript.

Kind regards,

Fabio Sallustio, PhD

Academic Editor

PLOS ONE

Journal Requirements:

Additional Staff Editor Comments (Senior Editor Joseph Donlan, jdonlan@plos.org):

Please note that the academic editor's request for additional data is not a strict requirement for publication in PLOS ONE. Providing the additional data would certainly increase the impact/interest of your work to the academic community, but if it is not possible to provide this data please simply resubmit your manuscript with a note in your response to reviewers that explains why you were unable to obtain/collect it.  
---

## [Author Response · Author response to Decision Letter 2]

6 Dec 2022

PONE-D-21-41101R3

Distribution of an analgesic palmitoylethanolamide and other N-acylethanolamines in human placental membranes

PLOS ONE

Reviewer #1: 

Please, consider, if possible, to add new data on PPAR-alpha.

Our reply:

We are working on the detection of PPARα receptor in amniotic membrane (AM), but since we want to publish data on PEA concentrations in AM preserved and stored under different conditions (cryopreserved, lyophilized, dry), we plan to accompany these results by images with immunohistochemically detected PPARα for each of technique used.

For this reason, we do not think it would be appropriate to show just incomplete data on PPARα in this paper. Additionally, PPARα is likely important for the analgesic effect particularly in the skin, which is target tissue of AM application. Its expression there has been already detected. Please see Results l. 328 – 332.

As an internal information, I include an image from the detection of PPARα in AM to this answer.

Image attached in PONE-D-21-41101R3_Response to Reviewers.

Immunohistochemistry for PARPα (green) in AM section. The nuclei are counterstained with propidium iodide (red).

It should be noted that the supplemental PPARα studies are quite demanding in time and resources and inclusion of even only partial results would require an extensive modification of some parts of the manuscript - Materials and Methods, Results, Discussion. 

On the other hand, should you insist on adding PPARα data we would accept it.

---

## [Editor Report · Decision Letter 3]

19 Dec 2022

Distribution of an analgesic palmitoylethanolamide and other N-acylethanolamines in human placental membranes

PONE-D-21-41101R3

Dear Dr. Jirsova,

We’re pleased to inform you that your manuscript has been judged scientifically suitable for publication and will be formally accepted for publication once it meets all outstanding technical requirements.

Kind regards,

Fabio Sallustio, PhD

Academic Editor

PLOS ONE
---

## [Editor Report · Acceptance letter]

27 Dec 2022

PONE-D-21-41101R3 

Distribution of an analgesic palmitoylethanolamide and other N-acylethanolamines in human placental membranes 

Dear Dr. Jirsova:

I'm pleased to inform you that your manuscript has been deemed suitable for publication in PLOS ONE. Congratulations! Your manuscript is now with our production department. 

Kind regards, 

on behalf of

Dr. Fabio Sallustio 

Academic Editor

PLOS ONE